# New Horizons or Business as Usual? New Zealand's Medico-Legal Response to Digital Harm

**Olivia Kelly**

School of Public Health and Interdisciplinary Studies, Auckland University of Technology, Auckland 1010, New Zealand; osimonekelly@gmail.com

**Abstract:** A socio-legal commentary, this article examines the emerging issue of digital harm in New Zealand's health settings. There are recent cases, and an increasing number of them, demonstrating the medico-legal response to various forms of digital harm. Of these, several representative cases are considered in order to identify features of digital harm within the health context. The article questions whether this is a *new* type of harm, enabled by the creation of new technologies, or simply a different manifestation of conventional unprofessional or unethical behaviour. The article considers whether the existing medico-legal framework can appropriately respond to this harm and whether new legal or policy tools are required. The cases suggest that the rights and disciplinary systems in place can adequately deal with digital harm within their existing scopes, particularly when individuals have been harmed. However, gaps in the legal framework are identified, with particular reference to the actions of unregistered providers and harm to professions. Further, a future challenge for the system may be the response to COVID-19 vaccine denial and misinformation. As the legal response to digital harm in the health context is a relatively unexamined area of research, this work may guide future research.

**Keywords:** digital harm; social media; professional discipline; unregistered providers

## 1. Introduction

The advent of the internet, and its associated technologies, has profoundly influenced all sectors of modern life, including healthcare. Digital technologies serve multiple functions in health care, including communications, access, consultation, administration, and increasingly, treatment. Although of immense benefit in many respects, one negative outcome is the ability to inflict harm through digital means.

New Zealand attempted to address this by creating a legislated framework to respond to and provide remedies for digital harm to individuals. This is the Harmful Digital Communications Act 2015 (HDCA) and applies generally across New Zealand society. The effect of the HDCA has been examined in specific contexts, attracting some criticism (King 2017; Hunt 2020).

This article narrows the focus, considering responses to digital harm within the New Zealand health context. In some cases, this emerging issue is faced by the relevant decision-making body for the first time.

The background to the medico-legal context is provided, with an exploration of how digital harm arises in this setting This includes an examination of the role and powers of the Health and Disability Commissioner (the Commissioner), the health rights watchdog, and the consequent entry into the judicial system for harmed consumers. The disciplinary pathway for registered professions is examined, with an exploration of the role and powers of the Health Practitioners Disciplinary Tribunal (HPDT). In New Zealand, this Tribunal is charged with making decisions about deregistration and other penalties, based on-among other grounds-professional misconduct and negligence.

Select cases are then analysed. There are relatively few cases dealing with digital harm in health environments and this article does not analyse all such New Zealand cases. The

article examines examples of harm or first-time considerations of particular technologies by the Commissioner or the HPDT. Privacy-based cases related to hacking or unauthorised accessing of health records have been excluded from this analysis, as the Privacy Act 2020 provides a comprehensive complaint pathway and remedies for these cases.

These early cases are used to consider the scope and forms of digital harm within the health and disability context. Two questions are addressed: Is the digital harm a 'novel' harmful behaviour? Or is it simply a different manifestation of unethical or inappropriate behaviour? The analysis examines whether the existing rights of consumers and disciplinary frameworks for health and disability providers are equipped to deal with digital harm, or whether health consumers are forced to look elsewhere for legal redress, including within the HDCA.

The cases are categorised, depending on the relationship between the provider causing the harm and the recipient of the harm. A service relationship, to which the Code of Health and Disability Services Consumer's Rights applies, may or may not exist.[1] The registration status of the provider is also relevant when the provider acts in a personal, rather than a professional, setting. Finally, some cases deal with harm to groups.

The resulting observations may offer preventive lessons for health providers in their use of digital technology. Gaps in the legal framework are identified from the cases, specifically in relation to unregistered professions and protections for harm to groups. The article then considers likely future issues, with COVID-19 vaccine denial and misinformation as the most relevant.

## 2. What Is Digital Harm?

The definition of digital harm is wide and seems to depend on the writer's focus. Kanungo et al. (2022) concluded that "having explored various effects of digital harm from different perspectives, the literature is rather fragmented and sparse in identifying the key constructs of digital harm and how to address them."

Clarity regarding the scope of 'digital' has emerged, as has some clarity around the notion of 'harm'. The concept of digital harm incorporates a wide array of use and misuse of *digital* technology. This can include—but is not limited to—harm caused by misuse of a variety of platforms, including social media, apps, text messages, or messaging services.

In 2012 the issue of digital harm was considered by New Zealand's Law Commission, an Independent Crown Entity tasked with law reform review (Law Commission 2012). The Law Commission identified unique features of digital harm that distinguish it from offline communication including instant dissemination, widespread accessibility, the persistence of digital information, and the ability for users to act anonymously. It may include cyberbullying behaviours, harassment, criminal activity, and other unwanted communication.

Further, the harm element may include "the full range of serious negative consequences which can result from offensive communication including physical fear, humiliation, mental and emotional distress."[2]

Within health contexts, digital harm definitions incorporate a range of behaviours and technology. Examples from the literature focus strongly on the misuse of social media. Practitioners are reminded of the dangers of blurring boundaries when using social media, and the ethical, regulatory, and legal issues that can arise from this blurring (Geraghty et al. 2021). The distraction of smartphone use during clinical practice has also been considered (Cho and Lee 2016), as has medical misinformation (Trethewey 2020). There is less of a focus in the literature on practitioner harm to consumers—whether inside or outside the service relationship—through misuse of digital technology.

---

1   Bailey v The Medical Council of New Zealand [2021] NZHC 3168.
2   Law Commission (2012). *Ministerial Briefing Paper*. [p. 8].

### 3. The New Zealand Response

New Zealand wrestled with a description of digital harm during the formation of the Harmful Digital Communications Act 2015 (HDCA). The Act was a response to growing concern about teen cyberbullying with some particularly serious cases from the United Kingdom and the United States. The Law Commission identified this as *new* behaviour to which, "existing law [was] not always easily applied."[3]

The Act describes digital harm in the interpretation section:

**"digital communication**—

(a)    means any form of electronic communication; and
(b)    includes any text message, writing, photograph, picture, recording, or other matter that is communicated electronically

**harm** means serious emotional distress".[4]

This wide definition of the mechanisms of digital harm aims to future-proof the system to allow for the inclusion of emerging digital technologies. The brief description of "harm" has been left to the Courts to further define, with an early interpretation subject to strong criticism from the High Court (Police v B [2017] NZHC 526).

Also, the HDCA's scope is defined in communication principles that outline the type of digital behaviour that may result in relevant harm. These principles include harassment, threatening, intimidating or menacing behaviour, sending obscene material, the release of sensitive information, making false allegations, matters in breach of confidence, incitement to harm or suicide, and denigration on various prohibited grounds.[5] The HDCA's criminal sections cover a range of very serious digital harm, including the dissemination of intimate visual recordings without consent ("revenge porn").[6]

The HDCA established a civil dispute resolution process through the appointment of an Approved Agency. Complaints must be sent to the Approved Agency, Netsafe. Netsafe's dispute resolution options include advice, negotiation, mediation, investigation, and conciliation. Once the Approved Agency process is completed, complainants have access to non-monetary court remedies.

The HDCA applies to harm done to individuals by other individuals or organisations. The HDCA's scope is not simply confined to public domains. Therefore, it incorporates conduct by health and disability providers. This also includes circumstances in which providers interact with consumers in their private lives, in the absence of any professional relationship.

### 4. New Zealand's Medico-Legal System

Parallel to the HDCA system runs the medico-legal system, of which two aspects are most relevant to digital harm. Firstly, New Zealand has an enforceable patient code of rights contained in the Code of Health and Disability Services Consumers' Rights ("the Code"). The Code contains ten rights and applies to all users of health and disability services. This is a deliberately wide scope. Although not an exhaustive list, the most relevant are the rights that cover respect, dignity and independence, freedom from discrimination, coercion, harassment, and various forms of exploitation. Finally, right 4 sets out a general right to services of an appropriate standard, including services that comply with legal professional, ethical, and other relevant standards.

Complaints about breaches of the Code are lodged with the Commissioner, an "independent watchdog" for the "promotion and protection of consumers' rights".[7] The Commissioner investigates a relatively small number of complaints about serious breaches

---

3    See (Ibid, p 14).
4    Harmful Digital Communications Act 2014, s 4.
5    See (Ibid, s 6).
6    See (Ibid ss 21–22B).
7    See Health and Disability Commissioner (n.d.).

of the Code and provides recommendary opinions for providers in breach (Manning 2009). A breach finding allows access to the judicial system via the Human Rights Review Tribunal (HRRT). HRRT remedies include damages to a maximum of NZD 350,000, although this maximum has never been awarded.

The Code and the Commissioner process applies to all providers of health and disability services. A smaller subset of this group is that of registered health practitioners, for which an added process for regulation and discipline also exists. The Health Practitioners Competence Assurance Act 2003 (HPCAA) enables registered professions to monitor practitioner competence and fitness to practice. The Act also provides for a single disciplinary body, in the form of the Health Practitioners Disciplinary Tribunal (HPDT). Section 100 of the HPCAA sets out the grounds for proceedings to be brought to the HPDT. These include professional misconduct, amounting to malpractice or negligence, including misconduct that may bring the profession into disrepute. The analysis below provides an example of where digital harm may qualify as professional misconduct. Other grounds relevant to digital harm include qualifying criminal convictions that reflect adversely on fitness to practice, and the practitioner acting outside their scope of practice. HPDT penalties include fines to a maximum of NZD 30,000, censure, conditions on practice, suspension, and deregistration.

The is a direct link from the HDCA to the grounds for HPDT proceedings. Criminal convictions under ss 22 or 22A of the HDCA may attract a term of imprisonment of up to 2 years, well over the qualifying timeframe of 3 months to trigger disciplinary proceedings under s 100(1)(c) of the HPCAA.

Code breaches may also result in disciplinary proceedings for the registered practitioner subcategory noted above, being brought to the HPDT by the Director of Proceedings, an independent Office with prosecutorial powers. In terms of digital harm, this is likely to be on the grounds of professional misconduct under s 100(1)(a) and/or (b) of the HPCAA.

It is clear, therefore, that there is a medico-legal system in place to deal with breaches of consumer rights by the broad category of health and disability providers, and the discipline of registered practitioners. The cases provide examples of whether, and if so how well, these systems address digital harm. The cases fall into specific categories depending on the existence and nature of a service relationship. The registration status of the provider is also relevant when no service relationship exists. Finally, some cases deal with harm to groups.[8] The exploration of harm to groups helps to assess the boundaries of the rights and disciplinary frameworks and to identify gaps.

## 5. Digital Harm within the Service Relationship: Non-Registered Provider

The first case involved text messaging within a service relationship. The Commissioner's opinion 09HDC01409 examined the services provided by a counsellor ("Ms C"), to her 18-year-old consumer with a mental illness. Counsellors—perhaps surprisingly—are not registered practitioners under the HPCAA.

The consumer had displayed suicidal thoughts, such that Ms C had given him a "no-suicide" contract. After a single appointment, the consumer cancelled several follow-up appointments. Despite this, when the consumer queried whether he could stop taking the anti-psychotic medications that his psychiatrist prescribed, Ms C replied via text, "I agree, no meds, but only if u have xlent support, at leart 2x wk, with therapist that r not afraid of emotional xpreshun. Txt me 2 make nxt appt when it wks 4 u. Blessings, [Ms C]."[9]

The consumer died by suicide two weeks later.

The consumer's mother complained to the Commissioner about the services provided by Ms C, claiming Ms C had breached right 4 of the Code (services of an appropriate standard). The Commissioner found multiple deficiencies in the care provided by Ms C.

---

8   For the purposes of this discussion, 'case' is used to encompass Commissioner recommendatory opinions and HPDT decisions.

9   09HDC01409, p. 7.

These included acting outside her scope of practice in providing advice on medications prescribed by a psychiatrist, and the lack of consultation with other providers.

The case also starkly highlighted the danger of texting as a method of communication within a counselling relationship. Texting is limited and subject to misunderstanding. It does not allow for the appropriate wrap-around service required for a vulnerable young person. The Commissioner commented:

> "While text messaging can be used appropriately when communicating with young people, there are recognised risks. These risks include lack of confidentiality, misinterpretation, and being "too available" . . . It was not appropriate for Ms C to provide advice to Mr A on medication, particularly by way of a text message."[10]

The Commissioner recognised that while many younger people preferred text messaging, as a communication method it had limited scope to deliver therapeutic mental health services in this situation. The Commissioner further recommended that text messaging be confined to simple task-based communication, such as making and confirming appointments.

The Commissioner found Ms C in breach of the Code and recommended that she review aspects of her practice and provide an apology to the consumer's family. The breach finding would have allowed proceedings to be brought to the HRRT, but it is unclear whether the family pursued this option.[11] Ms C would not have been subject to the HPCAA professional discipline regime, as counselling is not a registered occupation. This raises the legitimate question of whether this Code rights-based regime resulted in *real* consequences for the unregistered counsellor.

### 6. Digital Harm within the Service Relationship: Registered Provider

Text messaging and other unacceptable conduct have led to significant impacts on others in the health and disability setting. The following cases reveal the impacts on registered practitioners, who are subject to not only the Commissioner, but also the disciplinary HPDT regime.

A 2009 case from the HPDT also highlights the pitfalls of text messaging within the service relationship. Decision 211/Nur08/112D involved a mental health nurse's care management of a woman with a psychiatric illness.

The patient ("Ms BM") had a "history of an eating disorder, self-harming behaviour, depression, and [had] made a number of attempts at suicide by overdose."[12] It was agreed by her care team that Ms BM be able to contact the nurse by text message on his work mobile phone as required. After the care team's careful consideration, this method was deemed to be an appropriate method of communication with Ms BM. Shortly thereafter, Ms BM and the nurse began to exchange text messages.

A turning point in the service relationship began when the nurse provided his personal cell phone number to Ms BM on a day when the battery on his work phone was flat. The text communication then became more regular when there was no clinical need for it. A sexual relationship developed, parallel with the continuing clinical relationship. The final digital act came when, after the patient's clinical care had been transferred, the nurse sent an image of his genitalia to Ms BM via mobile phone.

The HPDT cancelled the nurse's registration and censured him. The nurse was ordered to pay a NZD 5,000 contribution towards costs and was fined NZD 500. The HPDT imposed conditions on the nurse's practice that, on any re-registration, he undertake post-graduate study in ethics and boundaries related to nursing, and work under supervision for a

---

[10]　See (Ibid, p. 13).

[11]　Any further proceedings in this case are not reported on the websites for the Commissioner or the HRRT. Private settlements may not be recorded on either site.

[12]　211/Nur08/112D, para 4.

period of 18 months. Before re-registration, the nurse would also be subject to medical and psychiatric examinations to ensure fitness to practice.

Clearly, the primary harm was the nurse's unethical breach regarding the sexual relationship, held to be "fundamentally unprofessional and inappropriate."[13] While not mitigating the clear ethical breaches involved, the immediacy and privacy created by text communication likely contributed to the relationship developing in this way. Arguably, the technology enabled poor ethical behavior by removing communication from any clinical oversight and providing the mechanism for the relationship to develop.

The provision of a personal mobile number was also relevant in a 2014 case from the Commissioner, 13HDC00733. In this case, a male general practitioner became enamored with a young female patient who had recently been discharged from a mental health service. As part of this discharge process, to increase her independence, the service stopped the patient's access to text-based support. The general practitioner, however, provided the patient with his personal mobile phone number, an action the Commissioner described as "clinically inappropriate."[14] In doing so, the general practitioner also bypassed the practice's text message system which would have allowed for clinical oversight of texts within his doctor-patient communications.

The general practitioner then went on to send the patient approximately 50 text messages in the subsequent 5–6 weeks. These were of an inappropriate nature, outlining his attraction to the patient, and attempting to organise non-clinical meetings. The patient consistently discouraged the text communication and the sentiments contained in the texts. She expressed confusion and discomfort about his role as a general practitioner in sending her such communications.

The Commissioner found that the conduct constituted harassment, thereby breaching right 2 of the Code. Further, the ethical transgressions breached right 4(2). The Commissioner referred the case to the Director of Proceedings, who initiated proceedings in the HPDT. Disciplinary penalties were imposed on the general practitioner, including a nine-month suspension, censure, conditions on his practice, and costs.

Arguably, this was a fully appropriate medico-legal response to this digital harm. The Commissioner's finding enabled the patient to pursue remedies for the breach of her rights should she wish to do so, and separate disciplinary penalties were imposed on the general practitioner. However, before the enactment of the HDCA (which also deals with harassment), it is unlikely that remedies under that Act would have added anything further to the outcomes in this case.

Also of note is that the general practitioner claimed that all his clinical consultations with the patient were "totally professional . . . directed towards the clinical problem that she presented with on the day."[15] This quote may indicate that he distinguished between his ethical obligations within his clinical office and the digital text space. In reference to a case with similar conduct, Basevi et al. (2014) noted that the case represented "how text messaging is not exempt from normal patient rights and maintenance of patient rights; they have real life consequences if not adhered to."

Again, this case serves as an example of ethical breaches enabled or exacerbated by the misuse of digital technology. It also serves as a reminder of the importance of applying ethical obligations to interactions beyond the physical and into the digital space.

## 7. The Service Relationship and Social Media

A newer technology, the use of social media via an app, was considered for the first time by the Commissioner in a 2021 case, 20HDC00906.

A 21-year-old nursing student ("Mr C") worked as a casual healthcare assistant in a rest home. Mr C took photos of two patients on his mobile phone and sent them to his

---

13　See (Ibid, para. 20.2).
14　13HDC00733, p. 4.
15　See (Ibid, p. 6).

"close friends list" on the Snapchat app.[16] One of the patients, with advanced dementia, was unable to consent to the photos being taken or disseminated.

Mr C's friends on Snapchat edited and captioned the photos and sent them back to Mr C. He then re-sent the edited versions back to Snapchat. One of the recipients, claiming to be offended on the patient's behalf, shared the photos on Facebook, and then with the media (Nightingale 2020).

The Commissioner found Mr C failed to treat the patient with respect and was therefore in breach of right 1(1) of the Code. He also breached right 3 by providing services in a way that did not respect the patient's dignity.

A noteworthy element of the digital harm, in this case, is the uncontrolled dissemination of the photos. Although the student shared the photos with a small group of friends, screenshotting and forwarding allowed for widespread dissemination, finally to a media site. This reflects the concept flagged in the Law Commission's 2012 report that once a digital image has been sent from a device, the initial sender's control over the image is lost.

The Commissioner referred to this phenomenon in its decision: "Mr C acknowledged the ease with which such photographs could be screenshotted, edited, and shared."[17] Further, "It was also short-sighted not to consider the possible wider consequences, given the ease with which information can be shared online."[18]

Although not a focus for the Commissioner, the case also shows the risk of mobile phone availability in the workplace. Phones provide immediate access to both a camera and the internet. Mr C seemingly acknowledged this when he agreed that in any future workplace he would not have his mobile phone on his person while working.

The Commissioner recommended that Mr C provide an apology to the patient's family and that he undertake a reflection exercise. The Commissioner stated that "Mr C has shown remorse for his actions and has learned a difficult lesson."[19] Further, however, the Commissioner referred this breach to the Director of Proceedings to assess whether any proceedings should be brought to the HRRT. As a mechanism normally reserved for very serious breaches of the Code, this case may serve as an early warning from the Commissioner about social media misuse.

Conceivably, the digital harm-focused HDCA could assist in this type of case should the patient or their representatives wish to have copies of images removed by website/app hosting services. Should the Approved Agency be unable to facilitate this, civil take-down orders are available from the District Court under s 19 of the HDCA. There is no equivalent capability within the medico-legal framework.

These cases of digital harm within an existing service relationship were addressed appropriately within the existing frameworks. It is clear that digital tools allow for misconduct to be enabled or exacerbated in new ways. Arguably, the frameworks are flexible enough, however, to capture and manage this misconduct.

## 8. Digital Harm outside the Service Relationship

The next issue is how the medico-legal system deals with harm outside an existing or prior service relationship. What can happen when a registered provider causes harm to an individual where no service relationship exists?

An instructive example is 300/Nur09/139P, a 2010 HPDT decision. This case involved a 58-year-old registered nurse ("Mr D"), who met a 14-year-old girl while she was staying in the Children's Ward at Rotorua Hospital. The girl had an intellectual disability and was described as "particularly vulnerable to the sexual advances of older men."[20]

Mr D met the girl in passing at the hospital and was not involved in her care. Nonetheless, he gave her his personal mobile phone number when they met in the hospital cafeteria.

---

16    20HDC00906, p. 3.
17    See (Ibid, p. 10).
18    ISee (bid, p. 11).
19    See (Ibid, p. 11).
20    300/Nur09/139P, para 26.

Over the next few months, Mr D called the girl and sent cards, presents, and text messages with sexual content. This continued despite his move to Saudi Arabia for work, and pleas from the girl's family to cease the communication.

Mr D set up a mechanism for communication over long distances by providing his mobile phone number. He also bought the girl a mobile phone, thereby bypassing parental oversight. The HPDT recognised that Mr D had made an "error of judgment" in providing his number to the girl, but that he had "compounded the situation by developing an inappropriate relationship in multiple respects."[21]

The HPDT dealt with the jurisdictional issue of the lack of service relationship by the application of s 100(1)(b) of the HPCAA. Mr D's actions constituted professional misconduct likely to bring discredit to the profession "that the health practitioner practised at the time that the conduct occurred." It was enough that Mr D was practising as a registered nurse at the time. He was not required to be in a service relationship with the girl.[22]

The HPDT cancelled Mr D's registration, censured him, and ordered costs of NZD 15,600.

Once again, the primary issue, in this case, was unethical grooming behaviour. This was exacerbated by the use of text messaging, particularly when the nurse was working overseas. The HPDT dealt very easily with this example of digital harm outside a service relationship.

Arguably, this application of s 100(1)(b) potentially allows for the discipline of registered practitioners in future digital harm cases, particularly around COVID-19 vaccine denial or misinformation.

It is clear, therefore, that the medico-legal framework is able to respond to digital harm from registered providers acting outside the service relationship. However, a gap exists for *unregistered* providers who cause digital harm to individuals with whom they are not in a service relationship. Paterson and Manning (2017) highlighted this long-standing gap in the medico-legal framework in dealing with the actions of unregistered providers, describing "the regulatory options in serious cases" as "limited and largely ineffective."

However, individuals harmed by unregistered providers can use the mechanisms under the HDCA, provided the provider's conduct breaches a communication principle or is an intentional criminal act. Of note, the HDCA does not enable the Approved Agency or the District Court to discipline providers or award damages to harmed consumers. Under s 19, the Court can order that an apology be published or that the defendants cease and refrain from the conduct concerned. These are untested powers but are unlikely to constitute a full response to digital harm complaints against unregistered health providers.

## 9. Harm to Groups

The next issue is the concept of harm to groups, rather than individuals. The Commissioner process, which deals with breaches of the rights of individuals, is excluded from this consideration. The question is whether groups can be harmed by the misuse of digital technology within a health context, and if so, how this harm is dealt with. It is also worth considering whether the medico-legal system should be used to remedy digital harm to wider, more amorphous groups, such as the general public.

The recent decision in 1114/Nur20/468P is illuminating. The HPDT made it clear that the actions covered by the case constituted a new type of issue, "namely the extent to which a health practitioner must avoid posting on social media comments that may be offensive to other members in their profession, health consumers and members of the public more generally."[23]

In 2019 the registered nurse posted comments on a public Facebook page accusing Taranaki-based Māori nurses of being, as summarised by the HPDT, "lazy, dishonest and unprofessional." Further, the nurse named specific workplaces, departments, and a specific

---

[21]　See (Ibid, para 56).
[22]　See (Ibid, para 35).
[23]　1114/Nur20/468P, para 1.

individual. This level of specificity potentially allowed for the identification of individual Māori nurses.

A witness, a registered nurse, described attending a 2-day hui (meeting) for Māori nursing students while the comments were posted. She reported the comments became a "significant distraction" at the hui, and many students were distressed and angered by the comments.[24]

This was a second strike for the nurse, having posted similar comments on Facebook in 2018. As a result of that conduct, she was required by her responsible authority to attend a cultural competence education programme, which she did not do.

As the 2020 decision was the first time the HPDT had considered harmful comments on social media from a practitioner, the Tribunal carefully considered how similar behaviour was dealt with in other jurisdictions. This examination incorporated several disciplinary cases regarding racist and discriminatory posts on social media. The cases considered how the posts brought their various professions into disrepute and harmed the public. The actions amounted to serious misconduct, also one of the grounds for discipline in the HPDT in New Zealand. The cases were therefore directly comparable and persuasive.

The nurse did not attempt to provide a freedom of expression rights-based defence, despite references to this in her original Facebook comments. She instead argued the speech was not racist as it was "true and based on her own past experience", an argument summarily dismissed by the HPDT.[25] A rights-based defence may have presented more of a challenge for the HPDT, and is likely to emerge in future cases.

The HPDT identified several different groups that were harmed by the nurse's conduct. The first was the individual nurses who may have been identified by the specific references to workplaces. Further, Māori nurses as a group were harmed, and well as Māori consumers "who are entitled to feel confident that they are not being dealt with by a nurse who hold racist views" were harmed.[26] Her colleagues and the identified professional nursing organisations were also harmed. Finally, the wider profession of nursing was harmed, given the comments were likely to bring, "substantial discredit to the nursing profession."[27]

The HPDT considered this to be professional misconduct. As a result of this and a second charge (which related to practising while suspended), the nurse's registration was cancelled with a two-year prohibition on an application for re-registration. Conditions including education and supervision were imposed on any future re-registration, and she was censured.

This served as a very clear message from the HPDT that there would be strong disciplinary consequences for registered health practitioners who harmed not simply individuals, but groups, via social media misuse. This was a response to a new type of harm enabled by the misuse of social media. However, although the HPDT took a comprehensive approach, the nurse's documented history of racism and intransigence made this a relatively straightforward decision.

The purpose of the HPDT's empowering Act is the protection of the public.[28] Combined with discredit to the profession as a part of professional misconduct, this clearly gives the HPDT scope to consider the effect of digital harm on groups. The real challenge for the HPDT in future cases may be when the potential harm relates to large amorphous groups, such as the general public, and if rights-based freedom of expression defences are utilised.

## 10. What Responses Are Missing?

The primary unaddressed issue in the current regime is the response to unregistered providers who cause digital harm to groups. The unregistered are not governed or disciplined by a professional body. Therefore, for example, racist digital communications would

---

24  See (Ibid, para 48).
25  See (Ibid, para 62).
26  See (Ibid, para 84).
27  See (Ibid, para 86).
28  Health Practitioners Competence Assurance Act 2003, s 3.

not result in deregistration for a counsellor, naturopath, or therapeutic masseur, despite bringing their professions into disrepute. This conduct also falls outside the jurisdiction of the Commissioner, which is focused on the rights of individual consumers, and the HDCA, which also deals with harm to individuals.

Arguably, this is one impetus for other occupations to become registered, to allow for discipline and to positively influence the public's view of the profession.

Also missing is the response to unregistered providers in the absence of a service relationship with a consumer. For the Code to apply and for the Commissioner to investigate conduct, services to the consumer must have commenced. A person subject to digital harm in this scenario may have to rely on other civil or criminal laws such as the Harassment Act 1997. The HDCA may also provide remedies, but these are not equivalent to HRRT or HPDT outcomes. There is no access to damages or disciplinary consequences for the unregistered provider.

## 11. Future Developments

New forms of digital harm emerged during the COVID-19 pandemic. Arguably, the types of harm included vaccine denial from providers, promotion of alternative unproven therapies (e.g., Ivermectin), and general COVID-related misinformation. The professional discipline of non-health professionals in relation to vaccine misinformation has been explored, noting the distinction between private and professional expression, and definitional challenges (Rychert et al. 2022). The harm within a health setting is arguably of a higher consequence, due to the public's reliance on health providers to provide accurate information.

Although there have been reports in the New Zealand media of multiple complaints to the Commissioner and the various HPCAA responsible authorities, cases have not yet emerged from either route (Broughton 2021).

For the registered, the response will likely include the disciplinary pathway under a professional misconduct charge, regardless of who has been harmed. The Commissioner may also address registered and non-registered providers if individual consumers have had rights breached, likely based on right 4(2) services to be provided that comply with legal, professional, ethical, and other relevant standards. For both the registered and non-registered providers, individual consumers might pursue a Code breach complaint, allowing damages to be awarded at the HRRT.

The HPDT, and any appeals to the High Court, are likely to include freedom of expression rights-based arguments under the New Zealand Bill of Rights Act 1990 (BORA), in a way that other digital harm highlighted in the other cases does not raise. Section 14 of BORA sets out the right to freedom of expression. Further, s 6 of BORA requires that when another law can be interpreted consistently with BORA, that meaning should be preferred. This BORA argument was introduced in early judicial review proceedings in 2021 (see footnote 1). The case was ultimately settled and discontinued before the substantive hearing. We await the next case to cite BORA grounds. This may present a substantial challenge to the scope of professional misconduct and the medico-legal disciplinary system.

## 12. Conclusions

These recent cases encompass several examples of egregious behaviour by health providers, including ethical boundary breaches, professional misconduct and negligence, and acting outside of their scope of practice. The use of digital technology has enabled and exacerbated the behaviour, allowing a broader scope of harm. But arguably, this is not wholly new or unique behaviour. The existing medico-legal system has been able to appropriately deal with the conduct within the existing rights and disciplinary pathways. Aggrieved consumers have not had to seek remedies under the legislation, the HDCA, which specifically deals with digital harm. This demonstrates the flexibility of the medico-legal complaints and discipline system.

The primary gap is the system's inability to regulate or discipline unregistered providers. This is a wider long-standing issue for the medico-legal system and requires political will for change and a legislative solution.

The cases also serve as reminders to health providers about the use of digital technology in practice, and the potential for the infliction of digital harm. The immediacy of the technology, the potential for uncontrolled dissemination, and the ability to facilitate inappropriate unmonitored communication are all potential hazards. These are well-documented hazards that apply within health and disability settings worldwide.

Future challenges for the system are likely to include disciplinary responses to digital harm from misinformation and vaccine denial-type conduct. The potential clash with rights-based freedom of expression arguments may prove a new challenge for the medico-legal system.

**Funding:** This research received no external funding.

**Institutional Review Board Statement:** Not applicable.

**Informed Consent Statement:** Not applicable.

**Data Availability Statement:** Data sharing not applicable. No new data were created or analysed in this study. Data sharing is not applicable to this article.

**Conflicts of Interest:** The author declares no conflict of interest.

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
