# Peer review of "New Horizons or Business as Usual? New Zealand’s Medico-Legal Response to Digital Harm"

_laws, 2023_

Round 1

Reviewer 1 Report

The article is an interesting Commentary on Digital Harm.

This topic is very interesting because it presents future prospects given the increase in digitization that we are witnessing.

Furthermore, the medico-legal aspects of Digital Harm have not yet been discussed in the literature.

I would recommend accepting this article because it starts from a general reasoning and takes advantage of the legislation in New Zealand that is advanced in terms of Digital Harm.

Since the article analyzes the medico-legal and socio-legal aspects of digitalisation, in the introduction it should also be noted that digitalisation is causing a change in the treatment scenario thanks to Digital Health Care and Telemedicine and that this change presents, in Healthcare, important Medicolegal Issues (You can for example use this reference: doi 10.3390/ijerph192315653)

Author Response

Thank you for your feedback. 

In regard to the treatment element, I have added a sentence in the introduction that refers to the increasing use of digital technologies for treatment purposes. 

Reviewer 2 Report

New Horizons or Business as Usual? Is a detailed and thoughtful commentary on an emerging issue in health care. The report is based on the New Zealand medico-legal framework. However, the issues raised are confronted by practitioners worldwide. 

The presentation of cases and the legal response are very well done.  The following are this reviewer's response to segments: 

1. Definition of digital harm. At the beginning harm was defined by the instrument (text, social media, etc) rather than the impact on the individual. This is a different approach that the usual definition of medical ethics or malpractice. Traditionally harm stems from the extent of the injury - illness, death, etc. Typically there is no distinction between injury caused by a surgical procedure and one caused by medication administration.  

The utility of HDCA may rest in the attempt to bring together mechanism of harm under a unitary legal framework that is perpetrated by health care professionals. It real value is that the injured do not need to comb through various categories of law - protection of minors, privacy, and slander. 

2. I am not familiar with New Zealand's Medico-legal code. The section describing it was very helpful. First, it is a national framework. In the US, elements of are state-based. This means that there are 56 different jurisdictions with their own legal frame.  Outlining the remedies available and the agency responsible for administering them were also enlightening. 

3. Line154 Has a typo or omitted sentence. 

4. The structure of sections 5 through 10 made a strong case for the creation of a national, digital harm law with a specific focus on health care.  The digital communications space is a new frontier for all of us. The authors' linkage to manifestations of bias (section 10) is particularly useful. In the US we are only beginning to explore the role of unconscious bias and stereotype threat in healthcare. I hadn't thought about the harm potention associated 'unregistered' providers. In the US, this group of providers are monitored through each state licensing agency. Perhaps this is a group of practioners that need some kind of oversight.  The writers clearly show that during the pandemic uncovered their potential for promoting bad medical information. 

Healthcare financing is the tie that binds oversight in the US and probably New Zealand as well. The rules are designed to insure that we are paying for ethical, quality care by knowledgeable practitions.  

The example of the Maori nurses are an example of the issues surrouding the drive to increase diversity among healthcare providers. Again, while none of us has the definitive solution, it is great that the writers bring the issues into sharp relief. 

Author Response

Thank you for your comprehensive and positive feedback. 

I have been unable to locate the typo or omitted sentence in line 154. Note after the addition of a new sentence, this is now line 155.